# Position: Adopting AI in Practice Does Not Guarantee the Productivity Boost

Won Ik Cho [1]   Seong-hun Kim [1]   Geunhye Kim [2]

## Abstract

This position paper argues that **adopting AI in organizational practice does not guarantee productivity gains, because human and environmental factors critically moderate the relationship between AI deployment and realized productivity improvements**. Following the advent of high-performance generative models, AI use has been rapidly encouraged in some sectors while being restricted in others. Most practitioners assume that AI brings productivity boosts owing to enhanced technical capabilities, but regardless of apparent performance advances in AI technology, human and environmental factors of the organization may substantially attenuate—or even negate—the effective productivity benefits. We identify five key moderating factors: human resource composition, baseline capability of individuals, learning curve of practitioners, incentives for fair use, and flexibility of objectives. Drawing on the partial equilibrium model of Gries and Naudé (2022), we argue that existing economic frameworks may inadvertently overlook these factors. We revise the existing framework to redefine effective organizational determinants and shed light on practical implications including industry and education, responding to alternative views and calling for action of stakeholders.

## 1. Introduction

Artificial intelligence transformation represents one of the most significant technological shifts across modern society, affecting laypeople, practitioners, and domain experts alike (Maslej et al., 2024). The applications of AI have expanded dramatically in recent years, encompassing search and information retrieval (Mitra & Craswell, 2018), recommendation systems (Zhang et al., 2019), personalized learning (Lin et al., 2023), creative content generation (Rombach et al., 2022), and task automation (Syed et al., 2020) across virtually every sector. Decision-makers in organizations increasingly assume that adopting AI will boost productivity, and this assumption appears theoretically sound if all users possess adequate understanding of what AI systems can and cannot accomplish (Bankins et al., 2024).

Recent research in economics has begun incorporating AI as a productivity factor in formal models of aggregate output. Graetz & Michaels (2018) provides empirical evidence on automation's effects on labor productivity across industrialized economies, while Acemoglu (2025) extends the task-based framework to compute AI's contribution to total factor productivity. These approaches make intuitive sense under the assumption that AI capabilities translate directly into organizational performance gains.

However, not all organizations achieve productivity improvements through AI introduction. A substantial body of evidence suggests that factors beyond technological capability determine whether AI adoption yields meaningful productivity gains. The "*productivity paradox*"—wherein AI capabilities advance rapidly while aggregate productivity growth remains sluggish—has been documented across multiple contexts (Brynjolfsson et al., 2019). Field experiments reveal substantial heterogeneity in AI's effects: Brynjolfsson et al. (2025) found productivity gains of up to 36% for the lowest-skilled workers, with minimal impact on the most experienced staff, while Dell'Acqua et al. (2023) demonstrated that for tasks outside AI's capability frontier, workers using AI performed 19 percentage points *less likely to be accurate* than those without AI access. Cross-country firm-level evidence shows that AI adoption is highly concentrated among large, already-productive firms, with complementary assets—including skilled workers, organizational processes, and data infrastructure—playing a critical moderating role (Calvino & Fontanelli, 2023). Acemoglu (2025) estimates that AI's contribution to total factor productivity may be substantially lower than commonly assumed, with effects highly dependent on the share of tasks that are easy- vs. hard-to-learn, and that whether feasibly automatable within the relevant horizon. The gap between AI's technical potential and realized organizational benefits echoes Solow's famous observation that "*you can see the computer age everywhere but in the productivity statistics*" (Solow, 1987).

[1]AI Center, Samsung Electronics, Suwon, Korea [2]Department of German, Hankuk University of Foreign Studies, Seoul, Korea. Correspondence to: Geunhye Kim <kghrav@gmail.com>.

*Proceedings of the 43rd International Conference on Machine Learning*, Seoul, South Korea. PMLR 306, 2026. Copyright 2026 by the author(s).

**We take the position that adopting AI in practice does not guarantee productivity boosts, because human and environmental factors—including organizational structure, individual capabilities, learning dynamics, usage incentives, and goal flexibility—critically affect the relationship between AI deployment and realized productivity gains. Traditional perspectives that treat productivity improvements as automatic consequences of AI capability adoption are thus insufficient and potentially lacking detail for practitioners.**

We identify five human and environmental factors that moderate AI productivity effects: (1) human resource composition of the organization, (2) baseline capability of individuals who interact with AI, (3) learning curve that practitioners experience, (4) incentives for fair and appropriate AI use, and (5) flexibility of objectives and key results. We argue that the economic model of Gries & Naudé (2022), while valuable, treats these factors as exogenous parameters rather than endogenous organizational variables that practitioners must actively manage. We will also see this productivity gap manifests differently across domains with high AI utilization, briefly investigating cases of industry and education.

## 2. Background

### 2.1. Recent Advances in High-Performance AI

The past several years have witnessed remarkable advances in AI capabilities, driven by a series of architectural and methodological breakthroughs. The scaling of autoregressive language models culminated in GPT-3 (Brown et al., 2020), which exhibited remarkable few-shot learning capabilities without task-specific fine-tuning. For coding, specialized models such as Codex (Chen et al., 2021) demonstrated the ability to generate functional code from natural language descriptions, powering tools like GitHub Copilot.

The introduction of instruction-tuning and reinforcement learning from human feedback (Ouyang et al., 2022) transformed language models into conversational assistants capable of following complex user instructions, exemplified by systems like ChatGPT. Most recently, agentic AI frameworks that integrate reasoning and action-taking (Yao et al., 2023) have extended language models to autonomous task completion through interaction with external tools and environments. These technical advances have collectively fueled widespread interest in AI adoption across sectors.

### 2.2. Human Perception and Productivity Research

Research at the intersection of human-computer interaction and computational social science has examined how humans perceive and interact with AI systems. Glikson & Woolley (2020) provides a comprehensive review of empirical research on human trust in AI, identifying factors including system transparency, reliability, and anthropomorphism that shape user attitudes and behaviors. Vereschak et al. (2021) surveys methodological approaches for studying trust in AI-assisted decision making, highlighting the complexity of human responses to AI assistance.

Empirical studies of AI's effects on work productivity have begun to emerge. Brynjolfsson et al. (2025) conducted a large-scale field experiment with customer support agents at a software company, finding that generative AI access increased productivity by 15% on average, with substantially larger gains (∼36%) for novice workers compared to experienced staff. Dell'Acqua et al. (2023) introduced the concept of a "*jagged technological frontier*," demonstrating through a randomized experiment with management consultants that AI improves performance for tasks within its capabilities while degrading performance for tasks outside that frontier. These findings suggest that productivity effects are highly contingent on task characteristics and worker attributes.

In education sector, Zawacki-Richter et al. (2019) systematically reviewed AI applications in higher education, finding that research has predominantly focused on technical implementations rather than pedagogical considerations. More recent work examining large language model adoption (Yan et al., 2024) identified practical and ethical challenges including academic integrity concerns, hallucination risks, and equity implications. Deng et al. (2025) conducted a meta-analysis of experimental studies on ChatGPT in education, finding positive effects on academic performance but no significant improvement in student self-efficacy.

### 2.3. Economic Models of AI and Productivity

The theoretical and economic modeling of AI's productivity effects has also drawn attention. The task-approach to labor markets, developed by Autor et al. (2003) and extended by Acemoglu & Autor (2011), provides a foundation for understanding how technologies affect the allocation of tasks between human workers and machines. Acemoglu & Restrepo (2018) incorporated this approach into an endogenous growth model, showing that automation has both displacement effects (substituting capital for labor in existing tasks) and reinstatement effects (creating new tasks in which labor has comparative advantage).

Gries & Naudé (2022) extended this framework by proposing a partial equilibrium model wherein AI provides *abilities*—rather than merely automating tasks—that combine with human skills to produce an aggregate intermediate service good. Their model specifies a human service production function for a unit interval $[N - 1, N]$:

$$H = \left( \int_{N-1}^{N} h(z)^{\frac{\sigma-1}{\sigma}} dz \right)^{\frac{\sigma}{\sigma-1}} \tag{1}$$

where $\sigma \in (0, \infty)$ is the elasticity of substitution (Sydsæter & Hammond, 2008), $z$ denotes tasks in the interval, and $h(z)$ is the output of task $z$. The output of tasks $z$ can be produced either by standard labor or by IT services where IT denotes technical assistance including AI:

$$h(z) = \begin{cases} \gamma_L(z)l(z)A_L & \text{if } z \in [N-1, N_{IT}] \\ \quad + \gamma_{IT}(z)b_{IT}(z)A_{IT}D \\ \gamma_L(z)l(z)A_L & \text{if } z \in (N_{IT}, N] \end{cases}$$

$$(2)$$

where $A_L$ and $A_{IT}$ represents general human skills and AI abilities (algorithms), $l(z)$ denotes the volume of hours employed in the task $z$, $b_{IT}(z)$ represents experts who apply AI to specific tasks, $D$ denotes data, and $\gamma$ terms capture task-specific productivity. It distinguishes between tasks below the automation threshold $N_{IT}$ (where AI can contribute) and tasks above it where only human labor applies.

The Gries-Naudé model demonstrates that the extent of AI adoption depends on: (i) relative abundance of AI capabilities compared to human skills; (ii) availability of AI-providing businesses and experts; and (iii) task-specific productivity of AI services relative to general labor. However, it treats these factors as given technological and economic parameters, without explicitly modeling the organizational, behavioral, and contextual factors that determine whether AI's potential productivity gains are actually realized.

# 3. Quantification of Human and Environmental Factors

We first propose five human and environmental factors that mediate the relationship between AI adoption and realized productivity gains, and see how these apply practically.

## 3.1. Human Resource Composition

The first factor concerns the organizational structure within which AI adoption occurs. Human resource composition indicates the hierarchy of the organization, specifically where AI policy managers, recipients of AI assistance, and practical supervisors are positioned relative to one another, and whether they operate in structural hierarchies or as peers.

This non-individual factor shapes how AI adoption decisions are made and implemented. The productivity impact of AI introduction is maximized when the human resource composition guarantees voluntary and informed use of AI (Venkatesh et al., 2003), where managers or peer systems effectively incentivize appropriate AI use, and where policy managers can adjust guidelines based on practitioner feedback regarding usability (Gritsenko & Wood, 2022).

In the Gries-Naudé framework, this factor affects the effective value of $b_{IT}$—the number of AI-providing experts—by determining whether organizational structures support or impede the deployment of AI expertise. An organization with hierarchical barriers between AI specialists and task performers may achieve low effective utilization despite nominally high $b_{IT}$.

## 3.2. Baseline Capability of Individuals

The second factor addresses the characteristics of individuals who receive AI assistance, particularly their understanding of AI principles and best practices, as well as their ability to perform tasks without AI support. This individual-level factor recognizes that AI assistance is not uniformly beneficial across skill levels.

As demonstrated by Brynjolfsson et al. (2025), novice workers may gain substantially more from AI assistance than experts. However, Dell'Acqua et al. (2023) found that for tasks outside AI's capabilities, reliance on AI assistance actually degraded performance, with the effect potentially larger for less experienced workers who cannot recognize when AI outputs are unreliable. The productivity impact of AI is maximized when individuals possess sufficient baseline capability to evaluate AI outputs critically, integrate AI assistance appropriately with their own judgment, and recognize the boundaries of AI reliability for their specific tasks. This factor best aligns with $A_L$ but is not completely independent with $A_{IT}$ since it concerns the human capability of appropriately and skillfully using AI.

## 3.3. Learning Curve of Practitioners

The third factor captures the temporal dynamics of AI adoption, specifically the time-gain trajectory individuals experience when first exposed to AI assistance for their work.

This human factor acknowledges that AI adoption involves learning costs. Practitioners must invest time to understand AI capabilities, develop effective prompting or interaction strategies, and integrate AI tools into existing workflows. The productivity impact of AI is maximized when the learning curve has low variance across individuals (avoiding Matthew effects (Gómez-Bengoechea & Jung, 2024) where those with initial advantages gain disproportionately), exhibits asymptotic convergence to effective usage, and does not depend strongly on baseline capability.

Significant heterogeneity in practitioners' learning curves can create organizational challenges, as some workers rapidly achieve productivity gains while others struggle, which may induce relatively unsatisfactory AI experience of practitioners or even perceived unfairness. This heterogeneity at first seems to relate only to the $A_L$ parameter in the Gries-Naudé model, as the human skills that enable effective learning about AI may be unevenly distributed. However, in global view, usual AI adoption deeply concerns the materials that are handled within the organization. Thus, this

should be considered in terms of (task-specific) AI abilities.

## 3.4. Flexibility of Objectives and Key Results

The fourth factor concerns how organizational goals align with the capabilities that AI assistance provides. Organizations set objectives that practitioners must achieve (so-called "performance indicator"), and these targets may or may not correspond to the performance improvements AI enables.

This environmental factor depends on organizational practices and the nature of work being evaluated. If an organization measures productivity in terms of outputs that AI can readily augment (e.g., code refactoring, document production), AI adoption may yield large measured productivity gains. If instead the organization values outputs that AI assists poorly (e.g., physical contribution), AI adoption may show minimal productivity benefit despite substantial effort (Damioli et al., 2021). This depends on domain – in software engineering for example, where AI is already human-level capable, expected performance of the organization is significantly stretched compared to pre-AI era. However, for manufacturing domain where current AI is weak, AI would not influence the overall goal of the organization.

In the Gries-Naudé framework, this factor relates to the task structure $[N-1, N]$ and the threshold $N_{IT}$ that separates automatable from non-automatable tasks. Organizations within the domain of inflexible objectives may effectively constrain the task set to regions where AI provides little advantage, limiting realized productivity gains regardless of AI's technical capabilities. However, with flexible goal setting, supervisors may reorganize the objectives to fully leverage the AI capability, which resultingly enforces the practitioners to adopt AI to meet the requirements (Dua, 2025). The productivity impact of AI is maximized when organizational objectives are flexible enough to incorporate AI-augmented work modes, when evaluation criteria appropriately recognize AI-assisted contribution as individuals' own, and when goal-setting processes can adapt as AI capabilities evolve.

## 3.5. Incentive for Fair Use

The last factor addresses the potential for inappropriate AI usage, including amplification of biases, propagation of hallucinated information, and inequitable evaluation outcomes that arise from differential AI access or capability.

Similarly to the organizational composition, this environmental factor depends on the nature of the work and the organizational context rather than individuals. In settings where AI misuse carries significant consequences—such as educational assessment, hiring decisions, or safety-critical applications—the productivity calculus must account for risks and safeguards. This factor is not explicitly repre-

sented in economic models of AI productivity but affects the feasible deployment strategies organizations can pursue. High-risk applications may require verification procedures, human oversight, or usage restrictions that constrain the labor-saving benefits AI could otherwise provide.

Different from the prior factors, the incentive (or reward) is difficult to define when it comes to fair use. Practically, adopting AI is the dominant strategy for individuals since it brings competitive results in shorter turn-around-time (Krakowski et al., 2023). However, it is fairly effective only when all participants agree to the similar degree of use of AI – which is usually not the case. Combined with the prior factor that regards both automatable and non-automatable tasks – especially if relative evaluation exists – such heterogeneity may resultingly bring inequality (Capraro et al., 2024), lessening the incentives of the group of individuals to develop their own skills. We claim that the incentive of using AI should be guaranteed only based on the homogeneity among practitioners, and the productivity impact of AI can be maximized if and only if incentives for fair use are strong, accompanied by monitoring mechanisms that detect misuse, given that the costs of unfair use are salient to practitioners. We augment this organizational factor so that it can influence the effective volume of AI expertise $b_{IT}$.

## 3.6. Revised Interpretation

The Gries-Naudé model specifies a human service production function wherein tasks $z$ are performed either with AI support (under $N_{IT}$) or with only human labor (Eqn. 2), where $\gamma_L(z)$ and $\gamma_{IT}(z)$ capture task-specific productivity of labor and AI respectively. This formulation treats the productivity parameters as exogenous technological givens. However, our analysis in this section demonstrates that realized productivity depends critically on human and environmental factors that mediate the relationship between nominal AI capabilities and actual performance gains. We therefore claim the need of an extended interpretation that addresses these factors explicitly.

### 3.6.1. ORGANIZATIONAL EFFECTIVENESS MODIFIER

We assume an organizational effectiveness factor $\Omega \in [0, 1]$ that captures how human resource composition and incentive structures affect the translation of AI capabilities into realized productivity. This modifier applies to:

$$\Omega = \omega_C \cdot \omega_I \tag{3}$$

where $\omega_C \in [0, 1]$ represents the *compositional alignment* factor, the degree to which organizational hierarchy facilitates effective deployment of AI expertise (Section 3.1), and $\omega_I \in [0, 1]$ represents the *incentive alignment* factor, the degree to which practitioners have appropriate incentives for fair and effective AI use (Section 3.5).

Consider some contrasting cases. In a flat AI task force where ML engineers, policy managers, and end users iterate on a shared goal, $\omega_C$ approaches unity; while in a multi-layer hierarchy where policy managers set goals for AI specialists that do not necessarily concern the productivity of task performers, $\omega_C$ collapses even when nominal $b_{IT}$ is high. Symmetrically, $\omega_I$ is high when all team members face matched access and incentives, and degrades when only a subset has reward for so-called "AI transformation", since the competitive asymmetry erodes peer incentives for fair use. In a hierarchical firm, $\Omega$ is therefore shaped chiefly by the layers between AI policy decision-makers and end users, whereas in an educational institution — where the school-teacher-student relationship is not strictly vertical — $\Omega$ varies more with task type than with hierarchical depth.

### 3.6.2. CAPABILITY-ADJUSTED PRODUCTIVITY

We extend the task-specific productivity parameters to account for baseline capability and learning dynamics. Let $\kappa_i \in [0, 1]$ denote individual $i$'s baseline capability (Section 3.2)—their ability to perform tasks without AI assistance and to critically evaluate AI outputs. The effective human productivity of the task becomes:

$$\tilde{\gamma}_L(z, \kappa_i) = \gamma_L(z) \cdot g(\kappa_i) \qquad (4)$$

where $g : [0, 1] \to \mathbb{R}^+$ is an increasing function reflecting that higher baseline capability enhances labor productivity.

For AI-assisted task-specific productivity, the relationship with baseline capability is non-monotonic, as suggested by the empirical findings of Dell'Acqua et al. (2023). Thus, we can think of a capability-task interaction function $\phi(z, \kappa_i)$ that is multiplied to $\gamma_{IT}(z)$. For tasks within AI's reliable capability boundary, $\phi$ may decrease with $\kappa_i$ (as in Brynjolfsson et al.'s finding that novices benefit more). For tasks outside AI's reliable boundary, $\phi$ may increase with $\kappa_i$ (as higher capability enables recognition of AI errors). Thus:

$$\phi(z, \kappa_i) = \begin{cases} \phi_{\text{in}}(\kappa_i) & \text{if } z \in [N - 1, N_R] \\ \phi_{\text{out}}(\kappa_i) & \text{if } z \in (N_R, N_{IT}] \end{cases} \qquad (5)$$

where $N_R \in [N - 1, N_{IT})$ denotes the threshold of tasks where AI outputs are reliable, $\frac{\partial \phi_{\text{in}}}{\partial \kappa_i} \leq 0$, and $\frac{\partial \phi_{\text{out}}}{\partial \kappa_i} > 0$.

### 3.6.3. TEMPORAL LEARNING DYNAMICS

The productivity evolves over time as practitioners learn to use AI effectively. Let $\tau$ denote time since new AI adoption. The learning curve factor $\lambda_i(\tau) \in [0, 1)$ captures individual $i$'s progress toward effective AI utilization:

$$\lambda_i(\tau) = 1 - e^{-\rho_i \tau} \qquad (6)$$

where $\rho_i > 0$ is individual $i$'s learning rate and the negative acceleration represents the trend after AI adoption (Pappas,

2026). Heterogeneity in $\rho_i$ across individuals creates the distributional challenges discussed in Section 3.3.

The time-varying effective AI productivity becomes:

$$\tilde{\gamma}_{IT}(z, \kappa_i, \tau) = \gamma_{IT}(z) \cdot \phi(z, \kappa_i) \cdot \lambda_i(\tau) \qquad (7)$$

For instance, practitioners with prior programming experience typically exhibit fast-converging $\lambda_i$ trajectories on coding-agent adoption, while those without such background show at first steep curves but larger variance across attempts. Thus, the learning rate may itself depend on individual's baseline capability $\rho_i = \rho(\kappa_i)$ and may have more a complicated relation. However, we deliberately leave this precise functional dependence in general form since committing it to a single closed formula would obscure the difference that comes from domain-specific structural gap.

### 3.6.4. ENDOGENOUS TASK BOUNDARY

The flexibility of objectives and key results affects which tasks organizations recognize as amenable to AI augmentation. We model this through an effective automation threshold $\tilde{N}_{IT}$ ($> N_R$) that may differ from the technical threshold $N_{IT}$ that assumes fully leveraging the AI capability:

$$\tilde{N}_{IT} = (1 - F) \cdot (N - 1) + F \cdot N_{IT} \qquad (8)$$

where $F \in [0, 1]$ represents organizational flexibility (Section 3.4). $F$ parameterizes under-adoption relative to Gries & Naudé's first-best $N_{IT}$; $F = 1$ (fully flexible objectives) indicates the original suggestion and $F < 1$ (rigid objectives) constrains AI application to a subset of performance-significant tasks, reducing $\tilde{N}_{IT}$ below $N_{IT}$.

### 3.6.5. REVISED PRODUCTION FUNCTION

Combining these modifications, we propose the following revised human service production function for individual $i$ at time $\tau$, where $\tau$ is is treated as a comparative-statics parameter; all equilibrium objects are evaluated at fixed $\tau$:

$$\tilde{h}_i(z, \tau) = \begin{cases} \tilde{\gamma}_L(z, \kappa_i) l_i(z) A_L & \text{if } z \in [N - 1, \tilde{N}_{IT}] \\ \quad + \tilde{\gamma}_{IT}(z, \kappa_i, \tau) \cdot \Omega \cdot b_{IT}(z) A_{IT} D \\ \tilde{\gamma}_L(z, \kappa_i) l_i(z) A_L & \text{if } z \in (\tilde{N}_{IT}, N] \end{cases} \qquad (9)$$

Aggregating across individuals and integrating over tasks yields the organizational-level human service:

$$\tilde{H}(\tau) = \left( \int_{N-1}^{N} \left( \sum_i \tilde{h}_i(z, \tau) \right)^{\frac{\sigma-1}{\sigma}} dz \right)^{\frac{\sigma}{\sigma-1}} \qquad (10)$$

Eqn. 9 aggregates over individuals within a single organizational subunit – a team, department, or working group

sharing objectives and working culture – not over an entire firm. Within such a subunit, the productivity distribution is far more tractable than the heavy-tailed firm-wide distribution suggested by anecdotes such as *"10× engineer."* Crucially, our formulation does *not* flatten individual heterogeneity: it is preserved through the baseline-capability parameter $\kappa_i$, the capability-task interaction $\phi(z, \kappa_i)$, and the learning rate $\rho_i$, so that outlier gains are accommodated rather than averaged out (Bass, 2014).

Our revised formulation preserves the structural insights of the original Gries-Naudé model while rendering them conditional on the human and environmental factors.

**Task-specific productivity ratio.** The original interpretation emphasizes $\gamma_{IT}(z)/\gamma_L(z)$—the relative productivity of AI-augmented vs. standard labor for task $z$. Our formulation *extends* this to an individual and time-varying ratio:

$$\frac{\tilde{\gamma}_{IT}(z, \kappa_i, \tau)}{\tilde{\gamma}_L(z, \kappa_i)} = \frac{\gamma_{IT}(z) \cdot \phi(z, \kappa_i) \cdot \lambda_i(\tau)}{\gamma_L(z) \cdot g(\kappa_i)} \quad (11)$$

which depends on the individual's $\kappa_i$ through both numerator and denominator, creating heterogeneous task-specific productivities across workers. For $\tilde{N}_{IT}$ and thus $N_R$ to be unambiguously set, let RHS of Eqn. 11 be monotonely increasing across $z$ for fixed $\kappa_i$ and $\tau$. For tasks within AI's reliable boundary ($z \in [N-1, N_R]$), the ratio may be higher for low-$\kappa$ individuals (novices benefit more from AI). For tasks outside this boundary ($z \in (N_R, \tilde{N}_{IT}]$), the ratio favors high-$\kappa$ individuals who can detect and correct AI errors. The temporal dependence through $\lambda_i(\tau)$ implies that the productivity ratio evolves during the adoption period, potentially with attenuating tangent as practitioners ascend the learning curve, assuming that $\kappa_i$ co-evolves with $\tau$.

**Availability of AI-providing experts.** The original model's $b_{IT}$ parameter captures the supply of IT experts who can deploy AI capabilities to specific tasks. Our formulation *reinterprets* this parameter through the organizational effectiveness modifier. The effective availability becomes:

$$\tilde{b}_{IT}(z) = \Omega \cdot b_{IT}(z) \quad (12)$$

where $\Omega$ captures how organizational composition affects the translation of nominal expert availability into realized deployment. An organization may have high $b_{IT}(z)$ (high number of AI specialists) but low $\tilde{b}_{IT}(z)$ if hierarchical barriers prevent these experts from reaching task performers, if policy managers cannot incorporate practitioner feedback, or if structural misalignment impedes coordination. This distinguishes between *headcount* of AI experts and their *effective reach* within the organization.

**Relative abundance of AI capabilities versus human skills.** In the original model, the ratio $A_{IT}/A_L$ determines

the comparative advantage of AI-augmented production. This interpretation *remains valid* in our formulation, but note that by fixing other infrastructural factors:

$$\frac{\tilde{A}_{IT}}{\tilde{A}_L} \propto \frac{\tilde{\gamma}_{IT}(z, \kappa_i, \tau)}{\tilde{\gamma}_L(z, \kappa_i)} \propto \frac{\phi(z, \kappa_i) \cdot \lambda_i(\tau)}{g(\kappa_i)} \quad (13)$$

The nominal technological ratio $A_{IT}/A_L$ is thus modulated by learning progress ($\lambda$) and the capability-task matching to baseline capability ratio ($\phi/g$). Two organizations with identical AI investments may exhibit different effective ratios depending on these mediating factors. Notably, even when $A_{IT}/A_L$ is high, the effective ratio may be low if $\lambda$ is small (impedance in learning progress) or $\phi/g$ is not significant (AI adoption does not benefit the expertise).

**Summary of modifications.** Table 1 summarizes how each original Gries-Naudé determinant is modified in our revised formulation.

*Table 1.* Relationship between original Gries-Naudé determinants and revised formulation.

| Original Determinant | Modification |
|---|---|
| $N_{IT}$ (automation threshold) | Effective threshold $\tilde{N}_{IT}$ depends on flexibility $F$ |
| $\gamma_{IT}(z)/\gamma_L(z)$ (task productivity, especially $N_{IT}$) | Individual ($\kappa_i$) and time-varying ($\tau$) |
| $b_{IT}$ (expert availability) | Effective availability $\tilde{b}_{IT} = \Omega \cdot b_{IT}$ |
| $A_{IT}/A_L$ (relative advantage) | Effective ratio modulated by $\lambda$, $\phi/g$ |

The original Gries-Naudé distributional predictions—that AI adoption favors tasks and sectors where the three determinants align favorably—remain qualitatively valid. However, our formulation reveals that these determinants are not purely technological or market-given but are themselves shaped by organizational choices regarding composition ($\omega_C$) and incentives ($\omega_I$), individual capability ($\kappa$) and learning curve ($\rho$), and objective flexibility ($F$). The productivity paradox may thus reflect not only slow diffusion of AI technology but also systematic failures to optimize these mediating factors, while the influence of each factor depends on the domain and the characteristics of the organization.

## 4. Alternative Views

We acknowledge several credible positions that oppose or qualify our argument.

### 4.1. The Technological Determinism View

One opposing position holds that sufficiently advanced AI will eventually overcome human and environmental barriers to productivity gains. Proponents of this view may argue that as AI systems become more capable, user-friendly, and

integrated into workflows, the moderating factors we identify will diminish in importance. Historical precedents such as electricity adoption—which required decades of factory reorganization before productivity benefits materialized—suggest that current limitations may be transitional rather than fundamental (Baily et al., 2025).

We respond that while AI capabilities will certainly advance, human and environmental factors are not merely transitional frictions but structural features of organizational contexts. Even highly capable AI requires human judgment about when and how to deploy it, and this leads to the issue of expertise and accountability. The jagged technological frontier (Dell'Acqua et al., 2023) is not simply a function of current AI limitations but reflects the fundamental challenge of matching AI capabilities to heterogeneous task requirements. AI capabilities expand but so does the frontier's complexity, requiring human discernment.

Even granting the determinist premise, our framework lets us read off what changes and what does not. As AI reliability rises, the threshold $N_R$ shifts upward, expanding the range $[N-1, N_R]$ over which novices benefit and shrinking the error-detection regime $(N_R, N_{IT}]$ in which high $\kappa_i$ is essential. The relative importance of $\phi_{\text{out}}(\kappa_i)$ thus declines. Simultaneously, with more tasks falling under AI augmentation, the binding constraints *shift* to $F$, since more of the task set is now automatable and rigid objectives leave more potential gain on the table, and to $\omega_I$, since asymmetric access produces larger competitive gaps when AI applies more broadly. Better AI therefore does not dissolve our moderators in terms of productivity; it redistributes them across $F$ and $\omega_I$ — which remain organizational, not only technological, choices.

### 4.2. The Measurement Problem View

Another opposing position suggests that apparent productivity failures reflect measurement inadequacy rather than genuine productivity shortfalls. Traditional productivity metrics may fail to capture AI's benefits, which could manifest as quality improvements, reduced cognitive burden, or expanded task scope rather than increased output volume (Brynjolfsson et al., 2021).

We acknowledge this concern and agree that measurement matters. However, our argument does not specify any particular productivity metric except for the human service production function (Eqn. 1). The human and environmental factors we identify affect realized benefits regardless of how those benefits are measured. An organization with poor human resource composition will fail to capture AI benefits whether those benefits are measured in output quantity, quality, or worker satisfaction. Improved measurement would make our argument more precise but would not eliminate the compensating role of human and environmental factors.

### 4.3. The Wage and Cost View

Another natural concern, related to recent work on AI and inequality (Paić & Serkin, 2025), is whether our framework adequately accounts for wages and labor costs. Following Gries & Naudé (2022, Section 3.2), the per-task marginal-productivity condition $p_h(z) \, A \, \gamma(z) = w$ applies analogously to standard labor and to AI-providing experts, where $w$ is the wage and $p_h(z)$ denotes the price for a task $z$. Under our revised formulation, this leads to a comparison of *effective* productivities: $g(\kappa_i) \, A_L \, \gamma_L(z)$ for standard labor versus $\Omega \, \phi(z, \kappa_i) \, \lambda_i(\tau) \, A_{IT} \, \gamma_{IT}(z)$ for AI-augmented production. Wage dynamics are therefore *embedded* in the productivity parameters that our five factors modify.

One might further ask whether the explicit cost of AI services — license, subscription fees, infrastructure for local deployment — should be incorporated. While legitimate, this is a separate cost-side variable that does not displace the human/environmental moderators we identify. Whether AI adoption yields realized productivity gains continues to depend on $\Omega$, $\kappa_i$, and $\rho_i$ that correspond with the discussed individual and organizational factors, rather than on the budget assigned for practitioners' AI access.

## 5. Practical Implications

### 5.1. Industry

Though we've focused on widely generalizable organizational factors, the primary use case that applies the framework would concern enterprises in the industry. AI adoption in firms has been studied regarding AI readiness (Jöhnk et al., 2021) and organizational factors (Chatterjee et al., 2021). Jöhnk et al. (2021) identifies five categories that organizations must address: strategic alignment, resources, knowledge, culture, and data, where each category highly relates to our framework, and Chatterjee et al. (2021) focuses on the context of manufacturing and production, where adopting AI is itself a costly process. In line with the prior empirical findings, we make the abstract factors explicit in the productivity framework. We want to recall the observation that AI adoption is beneficial if and only if the enterprise is prepared for it, accompanied by the understanding of cost and impact beyond quantitative metrics.

Besides, the five factors of our framework manifest differently across industry subgroups. In knowledge work settings, as studied by Dell'Acqua et al. (2023), the jagged technological frontier means that AI assistance improves productivity for some tasks while degrading it for others. The baseline capability and learning curve both become critical: workers must recognize which tasks benefit from AI's reliable capability frontier. Groups with inflexible evaluation criteria may inadvertently incentivize workers to use AI inappropriately for tasks where non-AI-related capabili-

ties should predominate, e.g., inappropriately encouraging bias-inducing data-driven approaches for automation where even data format is not standardized by domain experts.

In operational settings such as customer service and human resources management, the learning curve appears less constraining but the organizational factor nonetheless matters. Encouragement of AI adoption is itself an opportunity for a big productivity boost, but the incentivization for practical adoption may have to accompany the analogous reward, followed by the flexible moderation of objectives and policies.

## 5.2. Education

Educational settings present distinct manifestations of the five factors. Positing students as recipients of AI in learning (with competitive peers) and teachers as supervisors who should also develop AI adoption scenarios, the policy and observation regarding the use of AI in classroom raises fundamental questions about educational objectives and fair use of AI which differentiates the case from industrial settings where effective productivity dominates. This echoes Koedinger & Aleven (2016) where early implementations of intelligent tutoring systems demonstrated learning gains in controlled settings, but scaling these benefits required addressing teacher training, curriculum integration, and assessment alignment—factors corresponding to human resource composition, learning curve, and flexibility of objectives in our framework.

Human resource composition in education involves relationships among administrators who set AI policies, teachers who implement instruction, and students who use AI tools. Effective AI integration requires alignment across these levels. Schools where teachers lack training in AI capabilities cannot guide student usage effectively, potentially amplifying inequities if some students develop AI literacy through external resources while others do not.

What operate distinctly in education are baseline capability and learning curve. Students tend to develop tool usage skills rather than deploying existing expertise – AI assistance that substitutes for learning activities may improve immediate performance metrics while impeding long-term skill development (Al-Obaydi & Pikhart, 2025). This creates tension between short-term productivity (assignment completion) and long-term outcomes (capability building) that the flexibility of objectives factor must address. Educational institutions should either redesign assessments to remain meaningful with AI availability (e.g., reorganize the objectives so as to let adopting AI bring individual growth) or build curricula where encouraging the tool usage does not impede the growth of expertise.

Contemporary concerns about large language models in education (Yan et al., 2024) also highlight the issue of fair use.

Academic integrity challenges arise when AI can complete assignments designed to assess student learning. Deng et al. (2025) suggests that ChatGPT can improve academic performance, but the lack of effect on self-efficacy raises questions about whether measured learning gains reflect genuine capability development. Despite limited sample size, Kosmyna et al. (2025) demonstrated the use of AI which directly deters the objective of education. Institutional and classroom policies should foster the fair use of AI, not making students entirely relying on them for the sake of productivity.

## 6. Call to Action

**Who benefits from the AI-driven productivity boost?** A critical question our framework must address is for *whom* AI-induced productivity improvements ultimately serve. We have treated productivity as a property of the organization's human service output (Eqn. 1, 10), but this leaves the beneficiary structure implicit. The answer differs fundamentally between industrial and educational contexts.

In industry, the explicit and immediate beneficiaries appear to be employees themselves: AI tools reduce cognitive burden, accelerate task completion, and — when the moderators align favorably — yield measurable individual gains. Once an organization confirms that an AI-augmented workflow is feasible, however, the ultimate beneficiary of *sustained* productivity improvement often turns out to be the executives: workforce restructuring, headcount optimization, and output-per-cost improvements are managerial prerogatives enabled by demonstrated productivity gains. In contrast, in education the tension largely dissolves. The explicit recipients of AI assistance — students and, to some extent, teachers — are also the intended ultimate beneficiaries, since the objective is to enhance learning rather than to extract surplus labor value. *"Productivity for whom"* in education therefore reduces to whether measured productivity (assignment completion speed, test scores) reflects genuine capability development or merely superficial output inflation — a concern already addressed by $F$ and $\omega_I$ (Section 3.4, 3.5).

Upon these, our position implies specific actions for researchers, developers, practitioners, and policymakers.

### 6.1. For Researchers and Developers

**Anticipate downstream effects of capability disclosures.** Machine learning researchers and developers working on industry-level applications should bear in mind that their research outputs and products, upon disclosure or deployment, may directly affect practitioners within relevant organizations. Some practitioners may be forced to follow cutting-edge AI capabilities for the sake of productivity. Researchers should therefore anticipate how their disclosures affect working groups, and should communicate clearly how

productivity is likely to be affected, under which circumstances, and for which kinds of users, so that laypeople and decision-makers are not misled into assuming unconditional productivity gains. Concretely, capability releases should report not only benchmark scores but also the context under which the gains were measured — a disclosure norm aligned with the $\phi(z, \kappa_i)$ structure of Section 3.6.2.

**Build educational AI with pedagogical guardrails.** ML researchers and developers working on educational or academic applications should recognize that the purpose of AI in these contexts is not to enhance superficial, short-term productivity. Concretely, they should (i) collaborate with pedagogy and human-computer interaction experts so that systems incorporate architectural guardrails such as withholding direct solutions (Kazemitabaar et al., 2024), or leverage alignment strategies that prioritize pedagogical advantage (Liu et al., 2024; Dinucu-Jianu et al., 2025), since recent evidence shows that unrestricted generative AI access can harm learning outcomes even while improving immediate task performance (Bastani et al., 2025); and (ii) embed explicit self-explanation or reflection steps into the interaction flow, since prevalent generative AI may induce metacognitive offloading, where users offload monitoring and evaluation to the tool (Fan et al., 2025). AI should serve as a complement to, not a substitute for, genuine learning.

**Develop empirical measures of human and environmental factors.** The five factors we identify require operationalization for empirical testing. Thus, organizational behavior researchers should develop instruments to assess the suggested human and environmental factors, especially when those definitions depend on domain. Longitudinal studies tracking organizations through AI adoption would be particularly valuable. Besides, economic frameworks should be extended to treat human and environmental factors as endogenous variables rather than exogenous parameters. This would enable more realistic predictions of AI's productivity effects and inform policy interventions.

### 6.2. For Practitioners

**Assess organizational readiness before AI deployment.** Organizations should evaluate their standing before investing in AI adoption. Tools for organizational AI readiness assessment (Jöhnk et al., 2021) should be expanded to incorporate our framework. In industry, supervisors should comprehensively discern the practitioners' need and capabilities, and in education, institutions may devise the incentivization regarding students' fair use of technology and curricula that captures both tool-using capability and personal growth.

**Invest in complementary organizational changes.** AI deployment should be accompanied by investments in training (addressing baseline capability and learning curve), organi-

zational restructuring (addressing human resource composition), policy development (addressing fair use incentives), and goal-setting reform (addressing flexibility of objectives). Whether to focus on training expertise (long-term) or the outcome (short-term) would depend on the task and domain.

**Monitor productivity outcomes with appropriate attribution.** Organizations should track productivity changes following AI adoption and investigate the factors contributing to success or failure. Revised assessment plans that take into account the AI adoption should primarily be discussed before practitioners set their strategy individually. Productivity shortfalls should not automatically be attributed to AI inadequacy when organizational factors may be responsible.

### 6.3. For Policymakers

**Incorporate human factors into AI adoption guidelines.** Policy guidance on AI adoption, especially when led by a separate suborganization with AI governance, should address not only technological considerations but also the human and environmental factors that moderate productivity effects. Guidelines that promote AI adoption without attention to these factors may lead to unproductive and superficial transition and backlash against beneficial technologies.

**Support research on AI productivity moderators and develop sector-specific frameworks.** Funding agencies should prioritize research that investigates the conditions under which AI adoption succeeds or fails, not only those that advance AI capabilities. The manifestation of human and environmental factors differs across manufacturing, knowledge work, and education – sector-specific guidance should be carefully developed to address the particular challenges, not recklessly adopting generic criteria.

## 7. Conclusion

This position paper argues that adopting AI in practice does not guarantee productivity gains. By identifying five human and environmental factors—human resource composition, baseline capability of individuals, learning curve of practitioners, incentives for fair use, and flexibility of objectives and key results—that moderate the relationship between AI deployment and realized productivity improvements, we extend the partial equilibrium model of Gries & Naudé (2022) by highlighting factors that prior economic models treat as exogenous but that practitioners must actively manage.

So far, the machine learning community has devoted enormous effort to advancing AI capabilities, which seemed to derive the productivity boost directly from adoption. Our position is that equivalent attention must be paid to the conditions under which those capabilities translate into effective productivity gains, and preferably before organizational members are systematically obliged to adopt AI.

## Acknowledgements

We sincerely thank the four anonymous reviewers for their constructive comments, thoughtful feedback, and engaging discussions that strengthened this paper. We are also grateful to our colleagues and the working groups across our organization and institution for providing inspiration and motivation in shaping the development of our positional statements.

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
