# OpenReview forum: "Position: Adopting AI in Practice Does Not Guarantee the Productivity Boost"
_ICML.cc/2026/Position_Paper_Track — ICML 2026 Position Paper Track regular_

### Official Review · Reviewer_3ZHb · 2026-03-15

**Significance:** 2
**Argument Clarity:** 3
**Rating:** 4
**Confidence:** 3

**Questions:**

See points under weaknesses.

**Alternative Views Section:**

Yes

**Compliance With Llm Reviewing Policy A Conservative:**

Affirmed.

**Discussion Potential:**

2

**Final Justification:**

The initial submission had some problems, primarily about the scope of who the "productivity gains" are for. I raised some of these concerns in the review, which were not sufficiently addressed in the rebuttal. During the second round of discussions, the authors provided excerpts of new text that clarify these concerns to some extent.

I am raising my score to "borderline accept" based on the discussions.

**Paper Summary:**

The use of artificial intelligence (AI) in organizations is currently promoted extensively with the promise of boost in productivity. In this work, authors point out that adopting AI without considering the human and environmental factors will not yield these productivity boosts, but instead can even hamper productivity. To address this, the authors modify an economic equilibrium model for AI to include factors that can account for five different factors identified by the authors. They also present a call for action that proposes suggestions to mitigate the mismatch between use of AI and productivity improvements.

**Position:**

Yes

**Position In Title:**

Yes

**Related Work:**

3

**Strengths And Weaknesses:**

### Strengths

* The critique put forth in this paper on the unscrupulous use of AI with the promise of productivity boost is timely, and provides useful insights drawing upon economic theory.

* The five factors identified by the authors to characterise the gap between adoption of AI and productivity boost are reasonable. Particularly, the factors relating to incentives for fair use and flexibility of objectives are nuanced.

* Authors adjust an existing economic model to incorporate these factors; the scope of doing approaching this based on theoretical models offers a useful tool for addressing the challenges with AI adoption.

* The paper is well-structured and clearly written. Relevant background is included,  for example, the economic equilibrium model is detailed adequately.


### Weaknesses

I have some major concerns with this work. These primarily are due to the normative stance the authors are operating from which assumes AI adoption will yield productivity boost; just that it is currently not aligned well and it can be fixed. This ignores the larger questions of if AI should be used at all in all organizations, and the impact this can have on wages and inequality. I elaborate further below:

1. This position promotes the normative stance that AI induced improvement in productivity is somehow impending, and currently it is not correctly measured. The call to action suggests ways to improve the measurement of productivity due to AI without questioning the structures of who this productivity improvements will benefit?

2. A key point that is missing in this position is that of labour and wages. How is the impact of AI on wages taken into account in this model? Several works have questioned the role of AI in exacerbating economic inequality [1]; I think these are important alternative views that need to be addressed in this work.

3. The main theoretical contribution is built on the  Gries-Naudé equilibrium model by introducing additional modifiers. I have several concerns about trying to _quantify_ several abstract elements. For example, how does one go about measuring the organization effectiveness, $\omega_C$? The elaboration provided in Sec. 3.6.1 takes up an extremely simplistic view of an important factor identified by the authors. I have similar concerns for the other modifiers that aim to measure attributes like individual capability $\kappa_i$, and learning rate $\rho_i$. The authors do no reflect on how easy or difficult or useful it is to quantify these features.

4. I am also missing some empirical evidence for the revised model. How is this model applicable to a setting, and how is this better?

### References

[1] Paić G, Serkin L. The impact of artificial intelligence: from cognitive costs to global inequality. The European Physical Journal Special Topics. 2025 Aug;234(10):3045-50.

**Support:**

2

---

> ### Author Rebuttal · Authors · 2026-03-28
>
> We thank Reviewer 3ZHb for the thoughtful and detailed review. We appreciate the recognition of the timeliness of our critique, the reasonableness of the five identified factors, the usefulness of our economic model adjustment, and the clarity of our paper structure. We address each concern below.
>
> - "This position promotes the normative stance that AI induced improvement in productivity is somehow impending... without questioning the structures of who this productivity improvements will benefit."
>
> Thanks for pointing out the core issue. We want to clarify our stance is that AI productivity improvement is conditional on human and environmental factors, rather than impending. However, we acknowledge that we did not sufficiently address the question of for whom productivity improvements matter, especially in call to action. This varies across contexts: in firms, executives often frame AI benefits in terms of management efficiency and finance; in education, policymakers assume AI benefits both students and teachers. We will revise the paper to address how AI benefits should be considered differently across sectors, making explicit that "productivity for whom" is itself moderated by the organizational factors we identify.
>
> - "A key point that is missing in this position is that of labour and wages. How is the impact of AI on wages taken into account in this model?"
>
> We acknowledge that labor and wage implications are important and we appreciate the reviewer's reference to Paić & Serkin (2025). To resolve the reviewer’s concern, we note that we took a view that is in line with Gries-Naudé (2022): if the price p is produced from a task z with labor h(z) with reward(wage) w, then pγA=w for both human and AI labor (see 3.2 of Gries-Naudé), so that the cost-aspect benefit of adopting AI in practice can be yielded by investigating A_L γ_L / A_IT γ_IT. Since we did not explicitly discuss the relationship between the wage and other factors in our manuscript, we will add it in the formulation section and also describe how observing the dynamics of overall factors concern the wage parameter.
>
> - "I have several concerns about trying to quantify several abstract elements. For example, how does one go about measuring the organization effectiveness, Ω?"
>
> We appreciate this concern regarding the operationalization of our introduced modifiers. We want to note two points. First, considering that the original Gries-Naudé model also does not provide detailed quantification formulas for its core parameters (e.g., γ_L(z), γ_IT(z), A_L, A_IT), we note that the purpose of the formal framework—in both the original and our extension—is to provide structural relationships that clarify how factors interact, rather than to provide ready-to-use measurement instruments. Second, we left the quantification general because the appropriate operationalization differs fundamentally across organizational types. In a classic hierarchical firm, Ω would be shaped by the layers of management between AI policy decision-makers and end users (employees), while in an educational institution where the hierarchy between administrators, teachers, and students is not strictly vertical, Ω would be more trivial factor compared to other organizations, while the detail may differ whether the task belongs to simple assignments or ones that affect the academic integrity. These details also demonstrate the difficulty of quantifying the parameters, and it holds for the individual capability and learning rate as well. Despite this challenge, we will add a separate section that discusses these domain-specific considerations to help readers understand how the framework can be adapted to their contexts.
>
> - "I am also missing some empirical evidence for the revised model. How is this model applicable to a setting, and how is this better?"
>
> Though we acknowledge that empirical evidence would strengthen the paper, we focused on the primary contribution of this position paper as the argument and conceptual framework, not empirical findings. However, we agree that elaborating on how the framework applies to specific settings would improve the paper's practical value. In the camera-ready version, we will expand Sections 4.1 and 4.2 to provide more detailed illustrations of how the framework applies in industrial and educational contexts, including concrete scenarios that demonstrate the practical implications of each factor. We also note that our Call to Action (Section 6.1) explicitly calls for longitudinal studies and field experiments as future empirical work that our framework enables and motivates.
>
> We sincerely thank Reviewer 3ZHb for the constructive and thought-provoking feedback. Feasible improvements—including expanded discussion of beneficiaries, acknowledgment of wage dynamics, domain-specific quantification guidance, and more detailed practical illustrations—will be addressed in the camera-ready version.

---

> > ### Author Rebuttal · Reviewer_3ZHb · 2026-04-02
> >
> > I thank the reviewers for their response.
> >
> > The authors promise several changes in the revised version. However, I would like them to describe the specific changes for the first two points with the corresponding text they want to include in the revisions. I understand the space is limited but we can continue the discussion into the response form. Please provide concrete text for the first two points.

---

### Official Review · Reviewer_HWGg · 2026-03-19

**Significance:** 3
**Argument Clarity:** 4
**Rating:** 5
**Confidence:** 5

**Questions:**

See the weaknesses section.

**Alternative Views Section:**

Yes

**Compliance With Llm Reviewing Policy A Conservative:**

Affirmed.

**Discussion Potential:**

4

**Paper Summary:**

The paper argues, contrary to popular belief, that adopting AI in various organizations does not imply an increase in productivity, which they refer to as the productivity paradox. Through an economics angle (the partial equilibrium model of Grier and Naudes, 2022), they show that Solow's (1987) remark that "the computer ages everywhere but in productivity statistics" is still persistent 40 years later in the era of generative AI. The paper takes a nuanced stance, studying human resource composition, base capabilities of individuals in the org, learning curves of practitioners, and incentives for fair human use, and explain why, through each of these lenses, the productivity of huumans in various domains does not necessarily benefit from adopting AI. The paper ends with a C4A for researchers to do more longitudinal studies of incorporating AI in various domains/orgs, and to extend the Grier-Naudes framework to treat humans/environmental factors as endogenous random vars, and for practitioners to assess organizational readiness before AI deployment, and for policy makers to support AI productivity monitors.

**Position:**

Yes

**Position In Title:**

Yes

**Related Work:**

4

**Strengths And Weaknesses:**

The paper is well supported and contains a lot of well-argued evidence and reasoning. In lieu of the recent trends for software companies to adopt the use of coding/research agents, and finance companies to adopt finance agents, this is an important and relevant topic to the ICML community, which is likely to inspire discussion on the topic. The paper writing is remarkably clear, offering an economic perspective and a balanced take among several viewpoints. The primary weakness of the paper is that the alternative view, that modern AI is still faulty and is getting better, is somewhat underemphasized in the discussion. It's an important point: if productivity suffers due to the faultiness of the AI, then better AI should improve productivity, and it would be interesting to very clearly argue the ramifications of this through the Grier-Naudes framework.

**Support:**

4

---

> ### Author Rebuttal · Authors · 2026-03-28
>
> We are grateful to Reviewer HWGg for the thorough and supportive review. We deeply appreciate the recognition of our paper's evidence and reasoning, its relevance and timeliness to the ICML community, the clarity of writing, and the balanced perspective among viewpoints. We address the primary concern below.
>
> - "The alternative view, that modern AI is still faulty and is getting better, is somewhat underemphasized in the discussion. It's an important point: if productivity suffers due to the faultiness of the AI, then better AI should improve productivity, and it would be interesting to very clearly argue the ramifications of this through the Grier-Naudé framework."
>
> This is a critical point and one we have carefully considered. We acknowledge that the technological determinism view—that sufficiently advanced AI will eventually overcome organizational barriers—deserves more thorough treatment in our Alternative Views section. We would like to clarify our reasoning for the current treatment and how we plan to strengthen it.
> Our core argument is that the human and environmental factors we identify are not merely artifacts of current AI limitations but are structural features of how organizations adopt and integrate any powerful technology. Historical precedent supports this view: the adoption of computers, the internet, search engines, and productivity software (e.g., spreadsheets) all followed similar patterns where organizational, human, and contextual factors critically mediated the translation of technical capability into productivity gains. The productivity paradox that Solow (1987) identified for computers persisted well beyond the point where computing technology was demonstrably capable. We believe a generalizable framework should not be contingent on a specific stage of technological development but should apply broadly to automation technologies.
>
> That said, we agree that explicitly analyzing how improving AI capability interacts with our five factors through the Gries-Naudé framework would substantially strengthen the paper. For example, as AI reliability improves, the threshold N_R (where AI outputs are reliable) shifts, which changes the capability-task interaction function ϕ(z, κ_i) and potentially reduces the importance of baseline capability for error detection—but may simultaneously increase the importance of flexible objectives and fair use incentives as AI becomes applicable to a wider range of tasks. We will elaborate this analysis in the revised Alternative Views section to make the dynamic relationship between AI advancement and our moderating factors more explicit.
>
> We thank the reviewer again for this valuable suggestion. This elaboration will be incorporated into the camera-ready version.

---

> > ### Author Rebuttal · Reviewer_HWGg · 2026-04-02
> >
> > I thank the authors for their detailed comments.
> >
> > Upon further thought, I wonder if the paper looks at the right metrics? In some sense, average productivity may not be the right metric, since one would expect this distribution to be more heavy-tailed: there's a famous phrase in the coding-agents community about "10x'ing your 10x engineer" with Windsurf/Cursor/etc. While the average worker at an organization may remain equally (un)productive, it's more likely that the most productive engineer might become more productive. Do the authors agree? How does this affect the formulation of the model?
> >
> > Moreover, I do agree with the primary concerns of Reviewer VxQu that something actionable for ML researchers should also be added to the call for actions. Also, a superficial point, I would encourage the reviewers to re-organize some of the equations to fit within the margins of the paper.
> >
> > Thanks,
> > Reviewer

---

### Official Review · Reviewer_VxQu · 2026-03-24

**Significance:** 2
**Argument Clarity:** 3
**Rating:** 4
**Confidence:** 3

**Questions:**

1. Some of the five factors could be seen as overlapping. Human resource composition and incentives for fair use both concern organizational structure and policy; baseline capability and learning curve both concern individual skill trajectories. Would it be possible to either demonstrate empirical discriminant validity or consolidate the factors?

2. How should others use the formalism in their thinking? How would it help the stakeholders to whom the action items are addressed?

**Alternative Views Section:**

Yes

**Compliance With Llm Reviewing Policy A Conservative:**

Affirmed.

**Discussion Potential:**

3

**Final Justification:**

I appreciate the authors rebuttal comments and find that the proposed revisions are helpful in orienting the paper and clarifying its contribution.

**Paper Summary:**

The paper argues for a strong and clearly delineated position, namely that the common assumption that adopting AI will automatically lead to productivity gains is not warranted. Instead, practitioners should assume that whether AI leads to productivity gains depends on five factors - (1) who is already employed (2) how good are the people who interact with AI (3) how quickly they learn (4) whether fair and appropriate AI use is incentivized and (5) how flexible objectives are.

**Position:**

Yes

**Position In Title:**

Yes

**Related Work:**

4

**Strengths And Weaknesses:**

The paper is well written and clear.  It provides a helpful overview of recent work in this area, along with a new formalism that could structure thought on this topic.  It is based on an extensive review of the recent economics literature on this topic. It presents alternate views and engages with them appropriately.

My primary concern is that the position is not actionable for the ML community.  The Call to Action mentions actions for three groups, none of whom are machine learning practitioners.  The actions for researchers are three tasks best done by economists; the actions for practitioners are aimed at business leaders and managers across any industry; and the actions for policymakers are, appropriately, directed squarely at them. There is no concrete action for the machine learning community, or part of the community, to do as a result of reading this position paper.

The references also suggest that the paper is responding primarily to the economics literature.  While I think this is totally fine for a position paper, since ICML has carved this out as a more interdisciplinary space, it would be important to show how the conclusions reflect back on machine learning research or practice. Otherwise it feels like the paper would be a better fit for an economics or management science/operations research journal, as that is the community best positioned to take up its argument and respond.

**Support:**

3

---

> ### Author Rebuttal · Authors · 2026-03-28
>
> We thank Reviewer VxQu for the careful and detailed review, and for acknowledging the clarity of our writing, the helpfulness of our overview, the extensiveness of our literature review, and the appropriateness of our engagement with alternative views. The feedback raises important points that we address below.
>
> - "My primary concern is that the position is not actionable for the ML community. The Call to Action mentions actions for three groups, none of whom are machine learning practitioners."
>
> We appreciate this concern and understand its basis given the ICML venue. Nonetheless, we would like to provide two responses for this. First, the ICML Position Paper Track CFP explicitly states that position papers "adopt a meta-level perspective on the field of machine learning, with wider scope than any individual area," and lists topics including regulation, ethical considerations, and the beneficial impact of the community's work. Our paper addresses the intersection of AI deployment and organizational productivity—a topic that is interdisciplinary by nature, which does not strictly fall into one of the categories but has visible overlaps. Second, we respectfully note that the ML community extends beyond researchers developing algorithms. Industry ML researchers and practitioners—who constitute a significant portion of the ICML audience—are directly involved in deploying AI within organizations and face exactly the productivity challenges we describe. The actions we propose for "practitioners" (Section 6.2)—assessing organizational readiness, investing in complementary changes, and monitoring productivity outcomes—are directly relevant to ML engineers and data scientists who lead or support AI adoption initiatives. We also believe the paper is of significance to the Science, Technology, and Society (STS) perspective, which increasingly intersects with ICML discourse. Furthermore, we will revise the Call to Action to make the relevance to ML practitioners more explicit.
>
> - "Some of the five factors could be seen as overlapping. Human resource composition and incentives for fair use both concern organizational structure and policy; baseline capability and learning curve both concern individual skill trajectories. Would it be possible to either demonstrate empirical discriminant validity or consolidate the factors?"
>
> This is a fair observation, and we acknowledge that some factors share conceptual territory. However, we chose to keep them distinct because they operate at different levels and have different practical implications. Human resource composition concerns the structural positioning of roles (who makes AI policy, who receives AI assistance, who supervises), while incentives for fair use concern the reward structures that govern how AI is actually used within any given structure. An organization could have excellent structural alignment but poor incentive design, or vice versa. Similarly, baseline capability is a state variable (current skill level), while the learning curve is a dynamic process (trajectory of skill acquisition over time). Consolidating these would obscure important distinctions that practitioners need when diagnosing why AI adoption is not yielding expected productivity gains. That said, we acknowledge that establishing empirical discriminant validity would strengthen the framework, and we view this as an important direction for future empirical work.
>
> - "How should others use the formalism in their thinking? How would it help the stakeholders to whom the action items are addressed?"
>
> The primary target audience for our formalism is practitioners—organizational leaders, managers, and ML deployment teams—who are contemplating or currently undergoing AI adoption. AI adoption involves substantial costs (licensing, infrastructure, training, workflow redesign) and labor investment. Our framework provides a structured way to think about whether and under what conditions these investments will yield productivity returns. Rather than assuming productivity benefits are automatic, practitioners can use the five factors as a diagnostic checklist: Is the organizational structure aligned? Do individuals have sufficient baseline capability? Is the learning curve being managed? Are incentives appropriate? Are objectives flexible enough? The formalism in Section 3.6 makes this more precise by showing exactly how these factors enter the production function, enabling practitioners to identify which factors represent the binding constraints in their specific context.
>
> We sincerely appreciate Reviewer VxQu's constructive engagement. We will revise the Call to Action to make the relevance to ML practitioners more explicit and will strengthen the discussion of how the formalism serves as a practical diagnostic tool in the camera-ready version.

---

> > ### Author Rebuttal · Reviewer_VxQu · 2026-04-04
> >
> > I appreciate the authors thoughtful responses, which have partially addressed my concern. I also see the authors draft response to HWGg regarding the Call to Action, which I appreciate as a response to my first concern. I wonder if the second part of the proposed call to action could be made more specific. Suggesting that researchers do things appropriately is always good, but definitionally true; likewise ensuring that AI serves to enhance genuine learning is surely the goal of most good-faith actors already.  Is there a call to action for the ML community that specifically stems from the intellectual contribution of the paper?
> >
> > re: "Second, ML researchers and developers working on educational or academic applications should recognize that the purpose of AI in these contexts is not necessarily to enhance superficial, short-term productivity. AI tools in education should not replace or foreclose opportunities for users' educational or intellectual growth. Developers should therefore deploy such tools appropriately and notify users that they should adhere to principles of academic integrity, ensuring that AI serves as a complement to—rather than a substitute for—genuine learning.”"

---

### Official Review · Reviewer_2EXR · 2026-03-25

**Significance:** 2
**Argument Clarity:** 3
**Rating:** 4
**Confidence:** 3

**Questions:**

See comments above.

**Alternative Views Section:**

Yes

**Compliance With Llm Reviewing Policy A Conservative:**

Affirmed.

**Discussion Potential:**

3

**Final Justification:**

The further rebuttal addresses my concerns and the authors should include intuitive illustrations in the updated version. Therefore I have updated my rating to borderline accept.

**Paper Summary:**

This paper argues that AI adoption may not lead to productivity improvement. More specifically, the authors consider five human and environmental factors that are critical to productivity when adopting AI. The paper extends the framework of the existing model of human-AI productivity by explicitly incorporating these factors. Practical implications and call for actions are sufficiently discussed, as well as alternative views.

**Position:**

Yes

**Position In Title:**

Yes

**Related Work:**

3

**Strengths And Weaknesses:**

Strength:

- The paper discusses an important topic, which is related to human-AI interaction or collaboration in many different domains. It is relevant and important to ICML community.

- Related work has been discussed, especially for economic models of AI and productivity. Also call for actions and practical implications are sound and thorough.

Weakness:

- Even though the position is based on more fine-grained modeling based on Gries & Naude (2022), it is still based on the assumption and modeling of existing work, which weakens the significance and makes the contribution incremental.

- In the section of revised interpretation, many newly proposed functions are not detailed, which makes it less intuitive to show why these functions would model it perfectly. E.g. $\phi$ is not specific in (5) and it would help if more examples were given.

- Overall, I am not convinced by the current math formulation to support the identified factors, and it needs more evidence to numerically simulate the new model, justifying that they work well under some typical cases.

**Support:**

2

---

> ### Author Rebuttal · Authors · 2026-03-28
>
> We thank Reviewer 2EXR for recognizing the importance of the topic, the thoroughness of our related work discussion, and the soundness of our call to action and practical implications. We appreciate the constructive feedback and address each concern below.
>
> - "It is still based on the assumption and modeling of existing work, which weakens the significance and makes the contribution incremental."
>
> We appreciate this concern regarding the novelty of our contribution. However, we would like to respectfully note that position papers are ‘judged primarily on whether they present a compelling position that warrants greater exposure within the machine learning community,’ and are expected to ‘make an argument for a viewpoint or perspective about what should be done, in contrast to main track papers, which report on advances that have already been accomplished.’ Our contribution is not a novel economic model per se, but rather the identification and articulation of five human and environmental factors that moderate AI productivity effects—factors that existing frameworks treat as exogenous but that practitioners must actively manage. We hope this distinction can help the resolution of the reviewer’s concern.
>
> - "Many newly proposed functions are not detailed, which makes it less intuitive to show why these functions would model it perfectly. E.g. ϕ is not specific in (5) and it would help if more examples were given."
>
> We acknowledge that the functional forms introduced in Section 3.6 could benefit from more concrete illustration. Our deliberate choice was to leave certain functions (e.g., ϕ(z, κ_i) in Eq. 5) in general form because the specific functional shape would differ across organizations, sectors, and task types. For instance, the capability-task interaction in a software engineering firm would take a different form than in an educational institution, reflecting fundamentally different task structures and skill distributions. We acknowledge that providing domain-specific examples would strengthen the intuition behind these functions, and we will include concrete illustrative examples (e.g., specifying plausible functional forms for knowledge work and educational settings) in the camera-ready version.
>
> - "I am not convinced by the current math formulation to support the identified factors, and it needs more evidence to numerically simulate the new model, justifying that they work well under some typical cases."
>
> We understand the desire for numerical validation and agree that simulation studies would be a valuable complement to our framework. However, we believe this falls more naturally into future empirical work rather than the scope of this position paper. The primary aim of our paper is to argue that human and environmental factors critically moderate AI productivity effects and to provide a conceptual and formal language for incorporating these factors into existing models. Numerical simulation—which would require specifying organizational parameters, calibrating to specific sectors, and validating against empirical data—constitutes a substantial research agenda in its own right. We view this as an important direction for future work that our framework enables, and we note this explicitly in our Call to Action (Section 6.1) where we call for longitudinal empirical studies and factorial field experiments. We will clarify this distinction more explicitly in the revised version.
>
> We are grateful for the reviewer's thoughtful engagement. Improvements that are feasible within the revision scope—particularly the addition of concrete examples for the proposed functions—will be incorporated into the camera-ready version.

---

> > ### Author Rebuttal · Reviewer_2EXR · 2026-04-04
> >
> > Thanks for the clarification and I understand that the review rules are different from those of the main track. However, my remaining concern is that it sometimes seems to be hard to tell what the math modeling looks like. It is beneficial to illustrate the functions or key parameters intuitively when discussing their properties.

---

### Decision · Program_Chairs · 2026-04-30

**Decision:**

Accept (regular)

**Comment:**

This article's notable contribution concerns the argument that AI adoption does not automatically lead to productivity gains due to human and organizational constraints. The authors seek to analyze a notable area by extending economic models to incorporate these moderating factors. Reviewers found the position timely and relevant, though concerns were raised about formalization, empirical grounding, and call to action. The rebuttal clarifies the conceptual nature of the work and strengthens the presentation; given the consistent borderline-positive consensus, I recommend acceptance.